# Evaluation of Pulsed Electric Fields (PEF) Parameters in the Inactivation of *Anisakis* Larvae in Saline Solution and Hake Meat

**DOI:** 10.3390/foods12020264

**Published:** 2023-01-06

**Authors:** Vanesa Abad, Marta Alejandre, Elena Hernández-Fernández, Javier Raso, Guillermo Cebrián, Ignacio Álvarez-Lanzarote

**Affiliations:** Departamento de Producción Animal y Ciencia de los Alimentos, Tecnología de los Alimentos, Facultad de Veterinaria, Instituto Agroalimentario de Aragón (IA2), Universidad de Zaragoza, 50013 Zaragoza, Spain

**Keywords:** pulsed electric fields, parasite, viability, hake, quality

## Abstract

Larvae of the nematode family *Anisakidae* are capable of causing parasitic infections in humans associated with the consumption of fishery products, leading to intestinal syndromes and allergic reactions. *Anisakidae* larvae are widely distributed geographically, with rates of parasitism close to 100% in certain fish species. Methods need to be established for their inactivation and elimination, especially in fishery products that are to be consumed raw, pickled, or salted, or which have been insufficiently treated to kill the parasite. Many strategies are currently available (such as freezing and heat treatment), but further ones, such as pulsed electric fields (PEF), have hardly been investigated until now. This study focuses on the experimental evaluation of the efficacy of PEF in the inactivation of *Anisakis* spp. larvae in terms of electric field strength, specific energy, and pulse width, as well as on the evaluation of the quality of fish samples after PEF treatment. Results show that viability of *Anisakis* was highly dependent on field strength and specific energy. Pulse width exerted a considerable influence at the lowest field strengths tested (1 kV/cm). Central composite design helped to define a PEF treatment of 3 kV/cm and 50 kJ/kg as the one capable of inactivating almost 100% of *Anisakis* present in pieces of hake, while affecting the investigated quality parameters (moisture, water holding capacity, and cooking loss) to a lesser extent than freezing and thawing. These results show that PEF could serve as an alternative to traditional freezing processes for the inactivation of *Anisakis* in fish.

## 1. Introduction

*Anisakis simplex* is a nematode parasite belonging to the *Anisakidae* family. It causes the most common human parasitic infection associated with the consumption of fish products, called anisakiasis. *A. simplex* is capable of infesting marine animals, such as fish or cephalopods, during its L3 larval stage. Humans, when consuming fishery products with the live parasite, become accidental hosts, which leads them to suffer from gastrointestinal infection, with symptoms such as abdominal pain, nausea, vomiting, diarrhea, fever in some cases, and even allergic reactions [1].

In recent years, the prevalence of cases of anisakiasis has increased worldwide, with more than 90% of cases reported in Japan and the rest in Europe. Spain is the European country with the highest incidence of anisakiasis, with 8000 cases per year [2,3,4,5,6]. This is due to the high prevalence of *Anisakis* spp. larvae in marine fish. Between 25% and 80% of fish that reach the fish markets are infested with *Anisakis*, although the rate of infestation can be higher in species such as hake (*Merluccius merluccius*) [7]. It is, therefore, necessary to apply a series of measures designed to inactivate this parasite, thereby limiting its access to the human consumption chain.

Regulation (EC) No. 2074/2005 [8] establishes that fishery products destined to be consumed raw or virtually raw (including cold-smoked, salted, or pickled products) must be frozen for at least 24 h at a temperature of −20 °C or below, or for at least 15 h at a temperature of −35 °C or below. The disadvantage of the freezing/thawing technology is the deterioration of fish quality, because the formation of ice crystals during freezing and storage causes dripping and softening of the meat when thawed [9,10]. Due to this, new processing techniques are under research, such as high hydrostatic pressure (HHP), microwaves, ohmic heating, ozone, ionizing radiation, or the use of chemicals such as essential oils and others, apart from salt, acetic acid, and other marinades. Although these can be regarded as alternatives to traditional processes, they still affect the sensory quality of the product in one way or another [11,12,13,14].

The technology of pulsed electric fields (PEF) has hardly been investigated until now for the inactivation of *Anisakis*. Moreover, although the use of PEF is widespread for the nonthermal inactivation of microorganisms in liquid foods [15], this technology has hardly been applied for the inactivation of zoonotic parasites, except for specific investigations on *Echinococcus granulosus* or *Ascaris suum* [16,17].

PEF treatments consist of subjecting a product placed between two electrodes, usually immersed in an aqueous solution, to high-intensity electric fields (between 0.5 and 30 kV/cm) by applying intermittent pulses of short duration (of the order of microseconds) without increasing the product’s temperature. To the best of our knowledge, only one article has hitherto been published on the use of this technology to inactivate *A. simplex* [18]. The results obtained by those authors are very promising, as they demonstrated almost 100% inactivation of L3 larvae of *Anisakis* in mackerel (*Trachurus japonicus*) fillets without affecting the product’s quality. Nevertheless, the range of evaluated PEF conditions and the manner in which the treatments were applied in that study could be limited in terms of industrial application. Therefore, a more systematic study of the influence of different PEF parameters on the lethal efficacy of PEF technology on L3 *Anisakis* larvae under the greatest possible number of controlled conditions is required. Thus, in this investigation, *Anisakis* inactivation by means of PEF treatments of different electric field intensity, specific energy, and pulse width will be studied in water solution in order to define a mathematical equation that would enable larval viability after treatments to be described. In a second step, the equation will be validated in hake meat in order to define the most suitable PEF treatment for *Anisakis* inactivation and to evaluate the impact of PEF on hake meat quality.

## 2. Material and Methods

### 2.1. Extraction, Isolation, and Assessment of the Larval Viability of L3 Anisakis

The lethal efficacy of several different PEF treatments was evaluated on larvae extracted from the abdominal cavity of whole hake (*Merluccius merluccius*) supplied by Scanfisk Seafood S.L. (Zaragoza, Spain). The larvae extracted from the fish were deposited in 50 mL beakers containing salt water (0.85% NaCl) with an electrical conductivity of 1.3 ± 0.1 mS/cm and stored at 4 °C. The salt solution in which they were stored was changed every 48 h. Preliminary studies carried out in the laboratory showed that L3 larvae remained viable during 2 weeks of refrigerated storage in the indicated saline solution.

To determine the viability of the larvae, the mechanical stimulation technique recommended by EFSA [1] was applied, according to which L3 larvae are considered alive if they move when mechanically stimulated with forceps or a punch.

### 2.2. Treatment of Anisakis spp. Larvae in an Aqueous Solution with PEF

For PEF treatments, 10 isolated larvae were selected, previously evaluated for viability, and immersed in a saline solution with an electrical conductivity of 1.3 ± 0.1 mS/cm inside a PEF treatment chamber. The electrical conductivity of the saline solution was measured at room temperature with an electrical conductivity probe (Almemo FYA641LF series, Alhborn, Germany). The PEF treatment chamber consisted of two parallel stainless-steel electrodes of 4 cm height and 4 cm length, separated by a 3 cm gap.

The pulse generation equipment used in this study (EPULSUS-PM-10, 2 kW, Energy Pulse System, Lisbon, Portugal) was a generator capable of applying square wave pulses of variable width (from 1 to 200 µs) with a frequency of up to 200 Hz. Maximum output voltage and current were 10 kV and 180 A, respectively. Processing parameters (load voltage, pulse width, number of pulses, and frequency) were controlled by a touch screen provided with the equipment, along with an oscilloscope (Tektronix, TDS 220, Wilsonville, OR, USA). Applied voltage, amperage, and pulse duration were measured with a high-voltage probe (Tektronix, P6015A, Wilsonville, OR, USA) and a current probe (Stangenes Industries Inc. Palo Alto, CA, USA), respectively, connected to the oscilloscope. The PEF conditions investigated ranged from 1 to 3 kV/cm, 3 to 50 kJ/kg, applying pulses from 3 to 100 µs. All treatments were applied at a frequency of 10 Hz.

For each PEF treatment condition (different electric fields, specific energies, and pulse widths), batches of 10 larvae were treated and each PEF treatment was performed in triplicate. After the treatments, larvae were removed from the treatment chamber and the effect of the different PEF parameters (treatment time, electric field, pulse width, and total specific energy) on the survival of *Anisakis* spp. larvae was investigated by post-puncture motility after 3 h of incubation at 4 °C in saline solution with 0.85% NaCl. Nonsignificant differences were observed when evaluating viability after 24 h of incubation.

#### Experimental Design

Response surface methodology (RSM) was used to determine optimal PEF treatment conditions capable of achieving the greatest reduction in *Anisakis* viability when treated by PEF in saline solution. A central composite design (CCD) was constructed to investigate the effects of electric field strength (from 1 to 3 kV/cm), specific energy (from 3 to 50 kJ/kg), and pulse width (from 3 to 100 µs) on the viability of *Anisakis* after PEF treatments, expressed in percentage (%). Each point of the CCD was carried out in duplicate.

The data thereby obtained were modeled with the following second-order polynomial equation:(1)$$Y=\beta_{0}+\sum_{i=1}^{k}\beta_{i}X_{i}+\sum_{i=1}^{k}\beta_{ii}X_{i}{}^{2}+\sum_{i>j}^{k}\beta_{ij}X_{i}X_{j}$$
where *Y* is the response variable to be modeled, *X_i_* and *X_j_* are independent factors, *β*_0_ is the intercept, *β_i_* the linear coefficients, *β_ii_* the quadratic coefficients, *β_ij_* the cross-product coefficients, and *k* the total number of independent factors. A backward regression procedure was used to determine the parameters of the models. This procedure systematically removes all effects that are not significantly associated (*p* > 0.05) with the response until a model with an exclusively significant effect is obtained. The CCD and the corresponding analysis of the data were carried out using the Design-Expert 6.0.6 software package (Stat-Ease Inc., Minneapolis, MN, USA).

To validate the equation, a new set of PEF treatments in the range of the conditions under study for the different parameters (field strength, specific energy, and pulse width), applied in saline solution were determined. Experimental results were compared with the estimated viability of *Anisakis* after the PEF treatments defined by the equation. In order to determine the final equation’s accuracy, R^2^ and the root mean square error (RMSE) were applied [19].

### 2.3. Treatment of Anisakis spp. Larvae in Hake Meat with PEF

In order to validate the mathematical equation previously obtained to describe PEF treatments applied in saline solution, PEF treatments were applied to 4.5 cm × 2.5 cm portions of hake, in which 10 larvae of *Anisakis*, obtained from the abdominal cavity of whole hake (*Merluccius merluccius*) as previously described, were placed. The larvae were “artificially” located inside the musculature by filleting the hake piece in half, placing the larvae inside, and tying up the piece with string. Each piece of hake was treated by PEF in a treatment chamber of 6 cm height and 6 cm length, separated by 5 cm gap, and containing a saline solution of 1.3 ± 0.1 mS/cm.

A series of different PEF treatments were applied in the range of 1 to 3 kV/cm but of constant pulse width (30 µs) and specific energy (50 kJ/kg). After treatments, larvae were extracted from the hake fillet and their survival was determined as previously described. Each PEF treatment was evaluated at least in triplicate. The survivability of *Anisakis* in the samples of fish meat after the different PEF treatments was compared to estimated survivability calculated by the mathematical equation previously obtained in aqueous solution.

### 2.4. Evaluation of Fish Quality after PEF Treatments

For quality evaluation, hake (*Merluccius merluccius*) loins supplied by Scanfisk Seafood S.L. were used. These loins were free of bones and larvae. Due to limitations of the PEF system, pieces of 4.5 cm × 2.5 cm of hake were treated with pulses of 3 kV/cm, 50 kJ/kg, and 30 µs. To evaluate the possible impact of PEF treatments on fish quality, an indirect study of their effect was carried out by measuring distinct properties that can be related to quality of fish meat: moisture, water holding capacity (WHC), cooking loss (CL), and color. To compare results, the quality of untreated hake fillets (control) thawed at 4 °C after freezing at −18 °C for 48 h was evaluated. At least 5 samples of fish meat were used for each procedure. For each sample, all analyses were performed in triplicate.

#### 2.4.1. Moisture Content

Pieces of 4 ± 0.2 g of fish meat were placed in an oven at 105 °C for 24 h [20]. The moisture content was calculated using the following equation:(2)$$Moisture\;\%=\frac{P_{i}-P_{f}}{P_{i}}\;\times \;100$$
where *P_i_* is the initial weight of the sample (grams) and *P_f_* is the final weight of the sample.

#### 2.4.2. Water Holding Capacity (*WHC*)

Samples of 5.5 ± 0.2 g were wrapped in sterile gauze and placed in a Falcon tube with glass beads. The tubes were centrifuged at 1300 rpm for 15 min (MEGAFUGE 1.0 R, Kendro, Germany) and the samples were weighed after centrifugation [21]. *WHC* was determined based on the following equation:(3)$$WHC\;\%=100-\left(\frac{P_{i}-P_{f}}{P_{i}}\;\times \;100\right)$$
where *P_i_* and *P_f_* are the initial and final weight of the sample (grams), respectively.

#### 2.4.3. Cooking Loss (*CL*)

Samples of 2 ± 0.1 g were placed in test tubes and immersed in boiling water until a temperature of 75 °C was reached inside the sample. Subsequently, the sample was recovered, the surface water was removed with paper, and the sample was weighed [22]. The following equation was used to determine *CL*:(4)$$CL\;\%=\frac{P_{i}-P_{f}}{P_{i}}\;\times \;100$$
where *P_i_* and *P_f_* were the initial and final weight of the sample (grams), respectively.

#### 2.4.4. Color Measurement

A colorimeter (Minolta CM-2002 series, Osaka, Japan) (Figure 5.7) was used to determine color parameters. Results were expressed using the *L**, *a**, *b** CIE color space. Based on these values, total color change (Δ*E**) was estimated using the following equation [23]:(5)$$\Delta E^{*}=\sqrt{{\left(L_{0}-L^{*}\right)}^{2}+{\left(a_{0}-a^{*}\right)}^{2}+{\left(b_{0}-b^{*}\right)}^{2}}$$

Where *L*_0_, *a*_0_, and *b*_0_ represent the initial values of color for nontreated fish meat (control), and *L**, *a**, and *b** are the values following PEF or freezing/thawing processes. Ten measurements were made of each sample.

### 2.5. Statistical Analysis

In order to study whether there were statistically significant differences between the parameters investigated in the different studies, one-way ANOVA with Tukey post-test was applied using the GraphPad PRISM^®^ program (GraphPad Software, San Diego, CA, USA). Error bars in the figures correspond to the standard deviation of the mean. The existence of statistically significant differences was determined for a *p* value of 0.05, and such differences are indicated with different letters in the graphs.

## 3. Results and Discussion

### 3.1. PEF Survivability of Anisakis in Saline Solution

As an example, Figure 1 shows the percentage of survivors to PEF treatments of different electric field strengths (from 1 to 3 kV/cm), pulse widths (3 or 50 µs), and specific energies (40 and 50 kJ/kg). As observed, the lethal efficacy of PEF depended, above all, on applied field strength, with which it increased almost linearly. Pulse width and specific energy play an important role at low electric fields of 1–1.5 kV/cm: the higher those two parameters, the greater the lethality. For an electric field of 1 kV/cm, survival of 90% of the larvae at 50 kJ/kg applying pulses of 3 µs was reduced to 50% by increasing pulse width to 50 µs. At this electric field, the applied energy between 40 and 50 kJ/kg did not exert a significant influence either. At electric fields equal to or over 2 kV/cm, neither pulse width nor energy in the investigated ranges had any effect on the lethal efficiency of the treatments, thereby indicating the important role played by the electric field in terms of larval survivability. According to these results, 90–100% inactivation could be achieved by applying treatments of 3 kV/cm, 40–50 kJ/kg, and both 3 and 50 µs pulses.

These treatments could be the most appropriate ones to apply for maximum inactivation of *Anisakis* L3 larvae. However, in order to evaluate in further detail the impact of PEF parameters (electric field strength—E, specific energy—W, and pulse width—P) on the survivability of *Anisakis* when treated in saline solution, a central composite design was carried out. Table 1 shows the results from the different PEF combinations and Table 2 shows the corresponding polynomial equation obtained after multiple regression analyses. Table 2 also includes the coefficients for each parameter of the equation, their statistical significance (*p* value) and 95% confidence limits, and the equation’s goodness of fit in terms of R^2^, R^2^-adjusted, and RMSE. The equation accurately described the experimental results based on these parameters. Among the three investigated parameters, the interaction between field strength and specific energy (*E*W*) was the most important parameter affecting the survivability of *Anisakis*, thereby confirming that, by increasing both parameters, the percentage of survivors would decrease (negative coefficient of *E*W* interaction). However, since the square of the field strength (*E*^2^) and the interaction of field strength with the square of the specific energy (*E*W*^2^) also showed a great statistical significance, an optimum field strength and specific energy to achieve a certain level of *Anisakis* inactivation can be obtained, as will be shown below.

Prior to carrying out estimations based on the equation shown in Table 2 and in order to validate that equation, new experimental results regarding the survivability of *Anisakis* after PEF treatments, featuring conditions different than those used in Table 1 in terms of the range of field strength (1–3 kV/cm), specific energy (9–50 kJ/kg), and pulse width (3–100 µs), were obtained. Figure 2 shows the relationship between the new experimental results and those estimated with the equation shown in Table 2, thereby confirming the goodness of fit of the mathematical equation in predicting the survivability of *Anisakis* after PEF treatments in saline solution within the range of the estimated treatment conditions. Then, based on the equation, Figure 3 shows the influence of pulse width (Figure 3A), specific energy (Figure 3B), and field strength (Figure 3C) on the estimated survivability of *Anisakis* L3 larvae when applying pulses of 50 kJ/kg (Figure 3A,C) and 30 µs (Figure 3B) in saline solution. As observed, pulse width would have exerted a considerable degree of influence on the lethality of *Anisakis* at low field strengths but, at 3 kV/cm, lethality would not be affected (Figure 3A). On the other hand, maximum inactivation would be achieved at around 40–50 kJ/kg (Figure 3B), and the highest lethality would be obtained at 3 kV/cm independently of the other PEF treatment conditions (Figure 3C). Based on these predictions, a PEF treatment of 3 kV/cm and 50 kJ/kg would provide the best processing conditions to reduce the survivability of *Anisakis* treated in water.

### 3.2. PEF Survivability of Anisakis in Fish Meat

In order to evaluate if the inactivation observed in saline solution was similar to that achieved in fish meat, a new set of conditions was applied using pulses of 30 µs and 50 kJ/kg at field strengths ranging from 1 to 3 kV/cm for the treatment of pieces of hake artificially parasitized with *Anisakis* larva. Figure 4 shows the survivability of *Anisakis* after applying different PEF treatments (square points) to pieces of hake compared with the survivability estimated by the equation shown in Table 2 and obtained in saline solution (continued lines; the dotted lines correspond with 95% confidence intervals). As observed, the experimental data are aligned with the estimations, thereby indicating that, under the tested conditions, the equation developed on the basis of treating *Anisakis* in saline solution would prove useful in determining the viability of PEF-treated *Anisakis* in artificially parasitized pieces of hake. These results would confirm that a treatment of 3 kV/cm and 50 kJ/kg could well be an optimal PEF treatment for limiting the survivability of this parasite in fish meat.

The results obtained in this study are in agreement with the scarce data published in the literature when applying similar PEF treatment conditions. Thus, the only available study related to PEF inactivation of *Anisakis* in fish (horse mackerel) showed around 10–30% immobilization of parasites when applying exponential pulses of around 1.4 kV/cm and an estimated energy and pulse width of 48 kJ/kg and 75 µs [18]. This level of inactivation was similar to that observed in our current study, either estimated by the equation shown in Table 2 and Figure 3 or obtained in pieces of hake meat (Figure 4). No further comparisons can be made, since specific energies over 50 kJ/kg were obtained in the study by Onitsuka et al. [18] at only 1.4 kV/cm. The advantage of the present study is that lower energy levels were applied to reach almost complete inactivation of *Anisakis*, whereas Onitsuka et al. [18] required estimated energies of 150 to 200 kJ/kg for similar levels of inactivation, since they applied lower field strengths. Lower energetic requirements could be of interest from a practical standpoint. In a possible industrial application, as described for other products [15], fish would be treated in continuous mode, requiring an increase in frequency to ensure that the required PEF treatment could be applied in a matter of seconds. High-energy treatments could lead to a noticeable temperature increase in the product, which could affect the quality parameters of raw fish. In the previous study, where temperature was controlled by renewing water and limiting the pulse frequency, PEF treatments hardly affected the quality of fish, indicating that PEF as a standalone technique could represent an alternative to other technologies, such as freezing, which are generally applied to assure the inactivation of *Anisakis* [18].

### 3.3. Fish Quality after PEF Treatments

In order to evaluate the impact of PEF treatments on fish quality, moisture, water holding capacity (WHC), cooking loss (CL), and color of pieces of hake were determined after applying 30 µs pulses of 50 kJ/Kg at 3 kV/cm. To compare results, the quality parameters of hake meat that were frozen stored for 2 days, and then thawed were also evaluated. Figure 5 and Figure 6 show moisture content (Figure 5A), water holding capacity (Figure 5B), cooking loss (Figure 5C), and color (Figure 6) of PEF-treated pieces of hake, freeze/thawed ones, and fresh ones.

Moisture measurement is an essential parameter to ascertain water content in muscle before and after a treatment and, in this case, to assess whether the electroporation produced by PEF could interfere in subsequent results regarding WHC and CL. As shown in Figure 5A, PEF treatments did not affect water content, but the same parameter was significantly reduced in the case of the frozen specimens. In other words, the electroporation that could be produced in muscle cells would not be sufficiently significant to produce a loss of water as it does in frozen pieces.

WHC indicates the loss of the capacity of fish muscle to retain water after centrifugation, which gives an idea of the juiciness of the flesh [24]. As observed in Figure 5B, results were similar to the case of moisture, thereby showing that PEF treatments did not affect this parameter, although WHC was significantly reduced in frozen samples. In other words, the electroporation that could be produced by PEF in the muscle cells would not affect water retention capacity. Freeze/thawed samples show the greatest difference compared with control; in other words, in terms of WHC, the freezing/thawing process affected fish quality to a greater degree than PEF. This implies that the juiciness of fish treated with PEF would be more pronounced than that of frozen/thawed fish and similar to that of fresh samples.

Cooking loss indirectly measures the level of damage suffered by proteins after treatments and subsequent cooking [25]. When samples are subjected to high temperatures (75 °C), the affected proteins become denatured and are no longer able to retain the water they contained. As shown in Figure 5C, the behavior of hake samples was similar to that displayed under other parameters, although, in this case, the pulsed samples displayed a slightly higher loss of water but this was not statistically significant. The obtained results might be indicating a possible slight effect on the most sensitive proteins or on other compounds interacting with water, despite the final temperature of the fish meat after PEF treatments never exceeding 15 °C. More research is necessary to clarify this effect of PEF at low temperatures. In the case of the freeze/thawed samples, CL was even lower than fresh samples, probably due to the fact that part of the water had already been lost in the previous stages. If all the water loss from the period prior to freezing was considered, these samples’ drip losses would probably be at least comparable to those of the control.

Finally, Figure 6 shows the different CIE *L** *a** *b** parameters, as well as the Δ*E** parameter that allows for the calculation of the samples’ difference in color with respect to the fresh product. As can be seen, the samples treated by PEF were similar in these parameters with respect to the fresh product, with a Δ*E** value lower than 3, which would indicate that, if there were differences in the values of *L**, *a**, and *b**, they would not be perceived by the human eye [26]. However, in the case of the frozen/thawed samples, luminosity was higher, as well as the *b** value, thereby indicating a more yellowish, luminous coloration in this type of sample. These differences were visible to the human eye, as likewise reflected by the values obtained for the Δ*E** parameter. In any case, the results obtained for color should be interpreted with caution, since variability among batches and even among parts of the same fillet is usually very high, and this could considerably affect results.

Figure 5 and Figure 6 suggest that a PEF treatment of 3 kV/cm and 50 kJ/kg would not significantly affect the quality of hake meat compared to fresh samples, although somewhat higher losses of water would be noted when cooking. In general, PEF-treated samples would be of superior quality than frozen samples. These results regarding quality are in agreement with those of Onitsuka et al. [18]; even when higher PEF energy levels were tested, the samples retained a texture closer to that of the control than to that of the freeze/thawed product.

## 4. Conclusions

Lethality of *Anisakis* when applying PEF treatments was largely dependent on PEF parameters, such as field strength and specific energy, whereby pulse width becomes determinant when low field strengths were applied. In this paper, a mathematical equation was developed which accurately described the lethality of PEF treatments in the range of conditions under study, both in saline solution as well as in hake meat. The tested quality parameters indicated that PEF, as a treatment technology, could be a promising alternative to freezing for the inactivation of *Anisakis* without affecting fish meat quality. In any case, further research is necessary to evaluate an even wider range of PEF treatment conditions; moreover, tests could be carried out on naturally parasitized fish to confirm the potential of PEF as a new strategy for the inactivation of *Anisakis* in fishery products.

## Figures and Tables

**Figure 1 foods-12-00264-f001:**
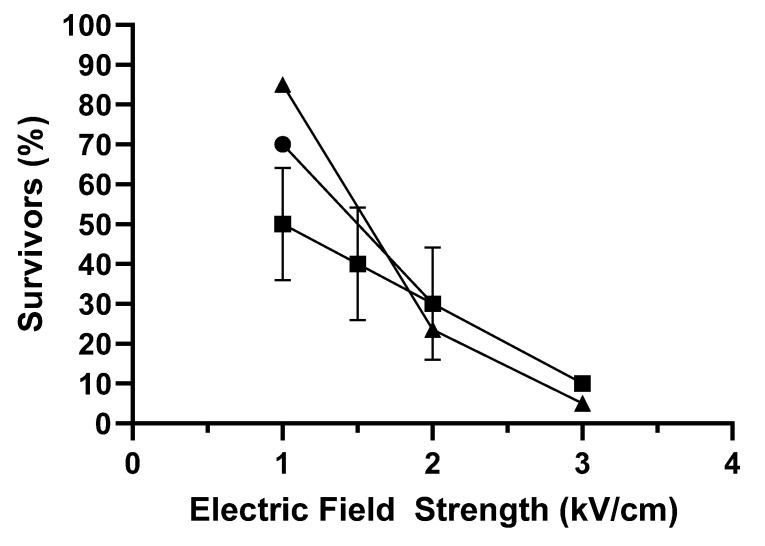
Influence of electric field strength on the percentage of survivors after 24 h of PEF treatments of different width and specific energy: 40 kJ/kg and 3 µs (●), 50 kJ/kg and 3 µs (▲), 50 kJ/kg and 50 µs (■).

**Figure 2 foods-12-00264-f002:**
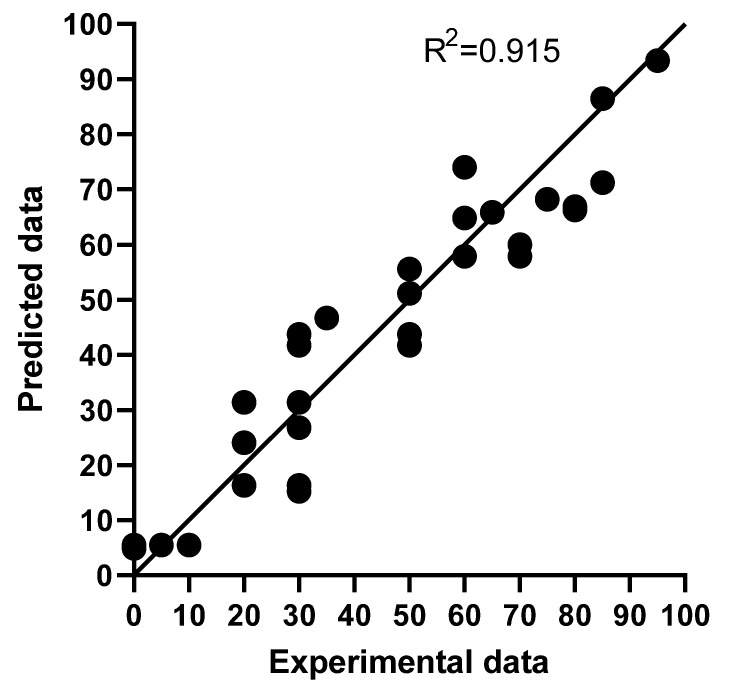
Relationship between the experimental survivability of *Anisakis* after the application in saline solution of PEF treatments of different rates of field strength (1–3 kV/cm), specific energy (9–50 kJ/kg), and pulse width (3–100 µs) compared with the survivability estimated with the equation shown in Table 2.

**Figure 3 foods-12-00264-f003:**
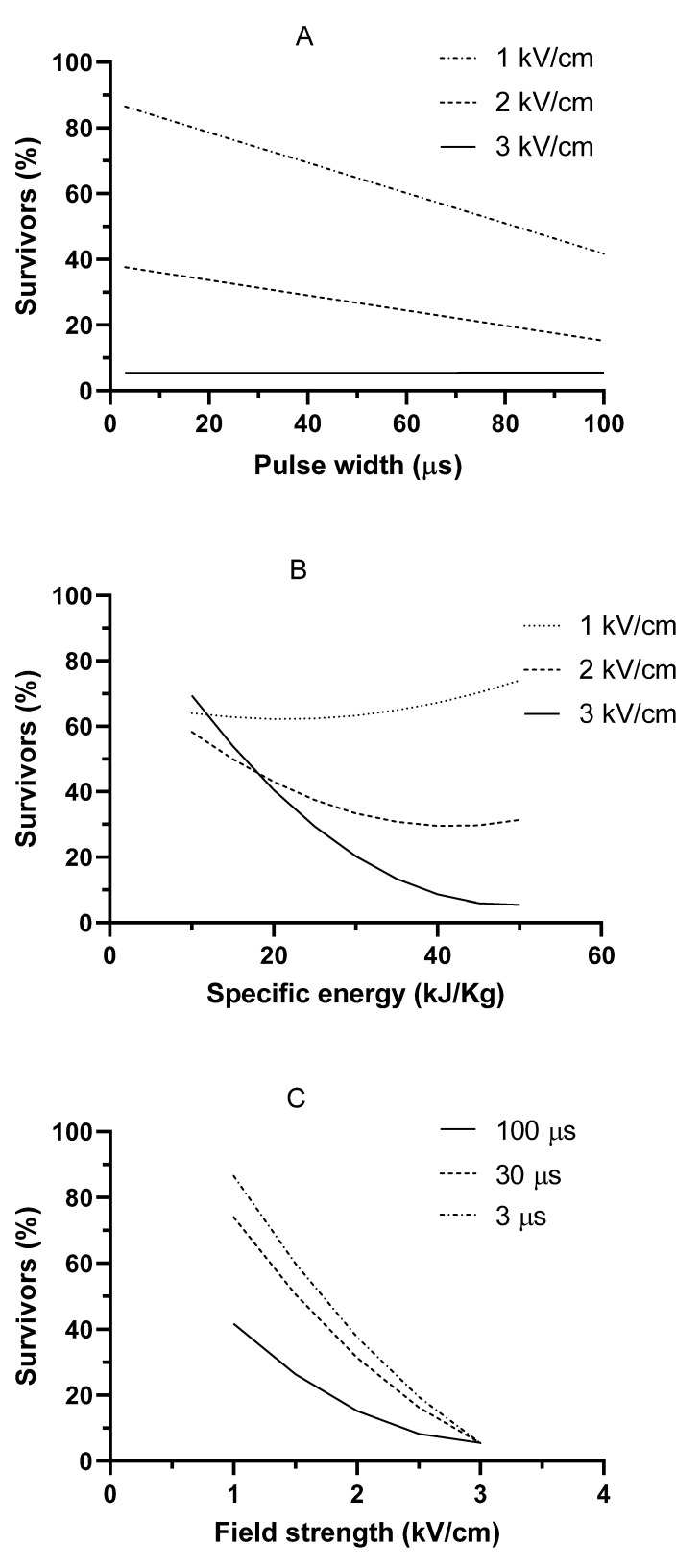
Influence of pulse width (**A**), specific energy (**B**), and field strength (**C**) on the estimated survivability of *Anisakis* larvae L3 when applying pulses of 50 kJ/kg (**A**,**C**) and 30 µs (**B**) in saline solution.

**Figure 4 foods-12-00264-f004:**
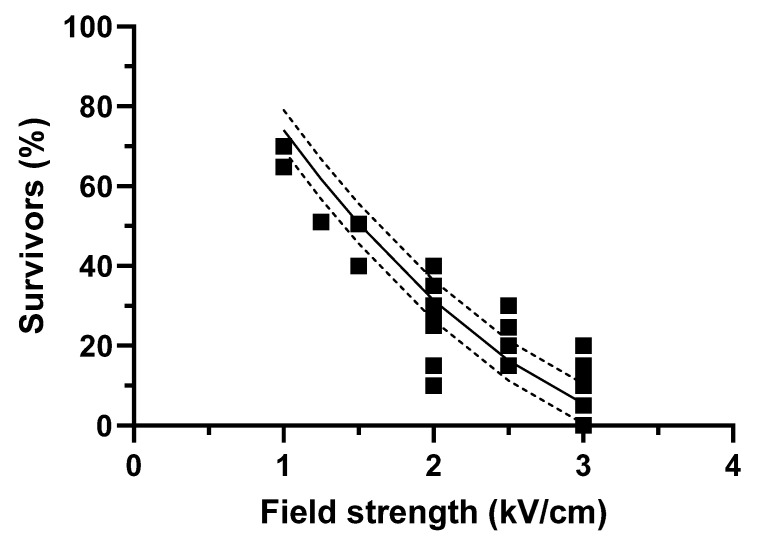
Survivability of *Anisakis* L3 larvae after applying 30 µs pulses of 50 kJ/kg at different field strengths (square points) to pieces of hake meat, compared with the survivability estimated (continued lines) by the equation obtained in saline solution shown in Table 2 (the dotted lines correspond to the 95% confidence intervals).

**Figure 5 foods-12-00264-f005:**
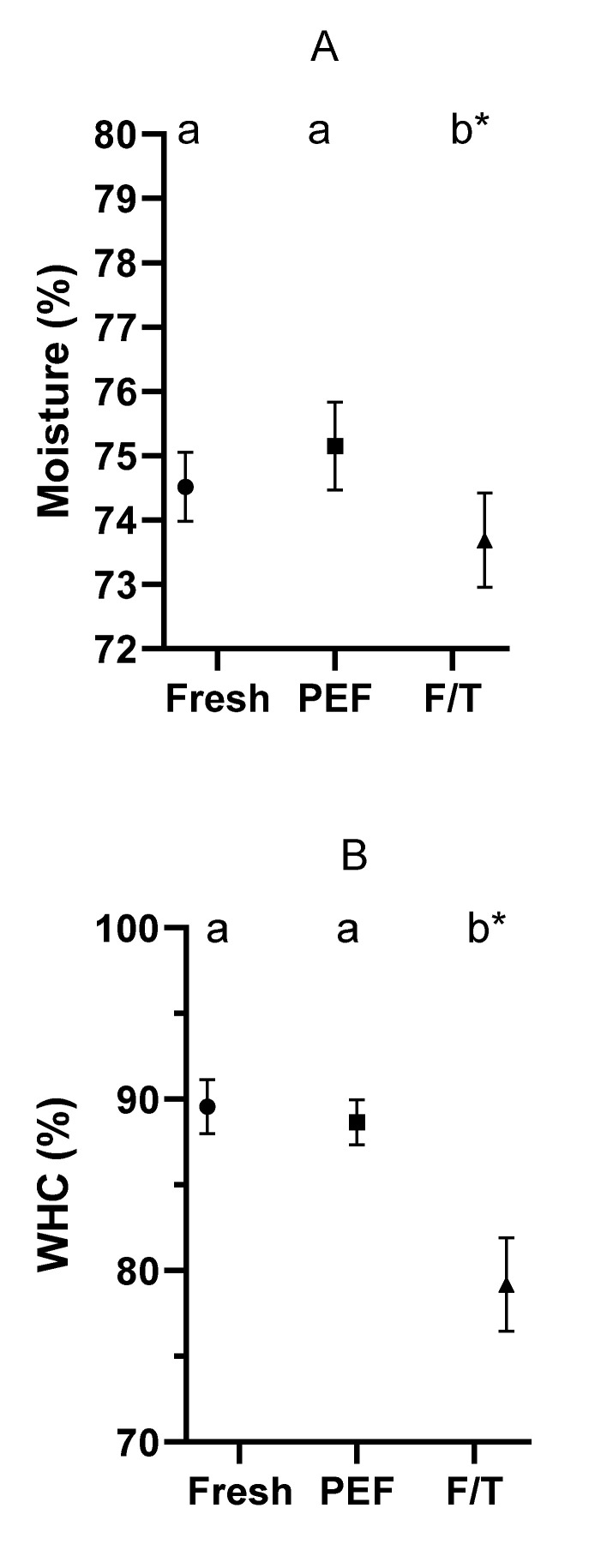
Moisture (**A**), water holding capacity (**B**), and cooking loss (**C**) of control (●), PEF-treated (3 kV/cm, 50 kJ/kg, and 30 µs) (■), and frozen/thawed (F/T) hake fillet samples (▲). Different letters indicate statistically significant differences among treatments (*p* = 0.05). The * indicates the statistical significance of the differences: * (*p* < 0.05); ** (*p* < 0.01); *** (*p* < 0.001).

**Figure 6 foods-12-00264-f006:**
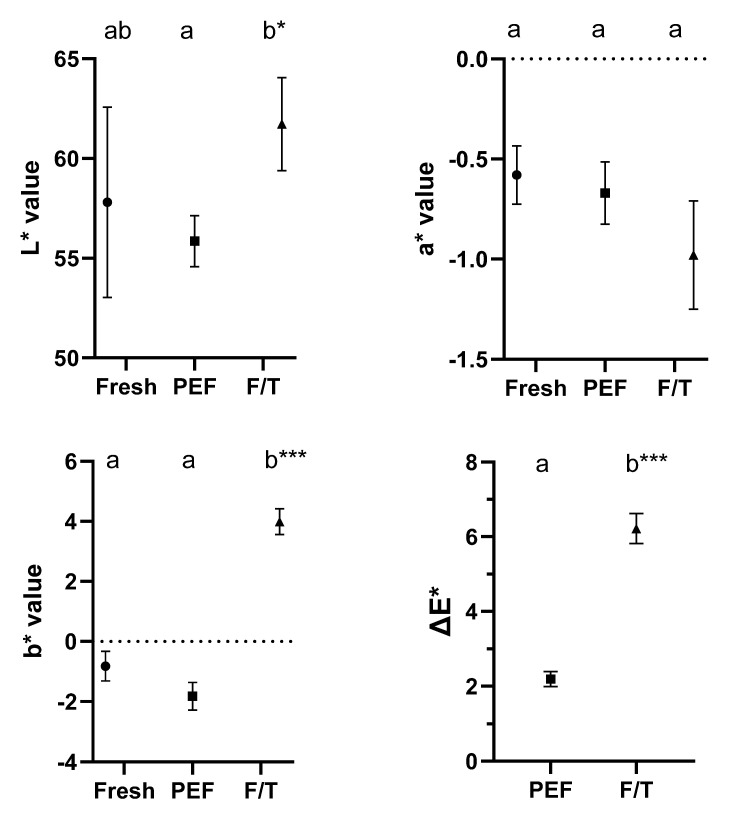
CIELAB parameters (*L**, *a**, and *b**) and total color change (Δ*E**) for control (●), PEF-treated (3 kV/cm, 50 kJ/kg, and 30 µs) (■), and frozen/thawed (F/T) hake fillet samples (▲). Different letters indicate statistically significant differences among treatments (*p* = 0.05). The * indicates the statistical significance of the differences: * (*p* < 0.05); ** (*p* < 0.01); *** (*p* < 0.001).

**Table 1 foods-12-00264-t001:** Central composite design evaluating the survivability of *Anisakis* L3 larvae after PEF treatments of varying field strength, pulse width, and specific energy.

Field Strength (kV/cm)	Specific Energy (kJ/kg)	Pulse Width (µs)	Survivability (%)
1	9	3	65
1	9	100	60
1	30	50	60
1	50	3	85
1	50	100	40
2	9	50	50
2	30	3	40
2	30	50	30
2	30	50	30
2	30	100	10
2	50	50	30
3	9	3	95
3	9	100	20
3	30	50	10
3	50	3	5
3	50	100	5

**Table 2 foods-12-00264-t002:** Polynomial equation describing the percentage of *Anisakis* survivability (*S*) treated in water solution after PEF treatments of different electric field strength (E), specific energy (W), and pulse width (P). Statistical significance (*p* value) and the 95% confidence limits of each parameter is included.

**S (%) = b_0_ + b_1_ × + b_2_ × P + b_3_ × E^2^ + b_4_ × E × W + b_5_ × E × P + b_6_ × W × P + b_7_ × E × W^2^ + b_8_ × E × W × P; R^2^ = 0.995; R^2^ Adjusted = 0.989; RMSE = 1.917**				
	Coefficient	*p* value	CL (−95%)	CL (+95%)
b_0_	59.29	1.402 × 10^−6^	49.95	68.63
b_1_	1.900	5.781 × 10^−6^	1.531	2.269
b_2_	0.515	0.000360	0.325	0.704
b_3_	8.388	2.213 × 10^−6^	6.975	9.802
b_4_	−2.210	1.613 × 10^−7^	−2.464	−1.956
b_5_	−0.482	2.943 × 10^−6^	−0.567	−0.397
b_6_	−0.02415	1.401 × 10^−5^	−0.02950	−0.01879
b_7_	0.01429	4.747 × 10^−5^	0.01047	0.01811
b_8_	0.01427	2.096 × 10^−6^	0.01188	0.01665

## Data Availability

Data is contained within the article.

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
