# Peer review of "Evaluation of Pulsed Electric Fields (PEF) Parameters in the Inactivation of Anisakis Larvae in Saline Solution and Hake Meat"

_foods, 2023, doi:10.3390/foods12020264_

Round 1

Reviewer 1 Report

In general, the study conducted by Abad et al, regarding the Evaluation of Pulsed Electric Fields (PEF) parameters in the inactivation of Anisakis larvae in saline solution is enough to provide innovations in the control of transmission of anisakis infection and can contribute to public health. However, I have several comments at some points:

1. I am interested in the method presented in inactivating L3 anisakis using pulse, but in the future, it needs to be reconfirmed how L3 that has been given pulse is really dead instead of half paralyze.

2. Economically, it is also necessary to understand whether this method is more efficient than freeze-thaw when applied on a large scale. (Need to address in the future)

3. I agree with the suggestion written by the author for further research on tests that could be carried out on naturally parasitized fish to confirm the potential of PEF as a new strategy for the inactivation of Anisakis in fishery products.

Author Response

In general, the study conducted by Abad et al, regarding the Evaluation of Pulsed Electric Fields (PEF) parameters in the inactivation of Anisakis larvae in saline solution is enough to provide innovations in the control of transmission of anisakis infection and can contribute to public health. However, I have several comments at some points:

  1. I am interested in the method presented in inactivating L3 anisakis using pulse, but in the future, it needs to be reconfirmed how L3 that has been given pulse is really dead instead of half paralyze.

Thank you very much for your comments that have been very useful to improve the quality of the manuscript.

Mobility test is the standard method to evaluate the viability of the L3 after a treatment. Although results of the mobility have been shown after 3 hours of the PEF treatment, it has been evaluated again after 24 hours of storage in saline solution and no differences have been observed concerning data obtained after 3 hours. This has been indicated in the M&M section.

  1. Economically, it is also necessary to understand whether this method is more efficient than freeze-thaw when applied on a large scale. (Need to address in the future).

We agree that this a point that needs additional evaluation in the future as it is indicated by the reviewer. At this moment the PEF technology is ready for large-scale treatments (i.e. potatoes are treated at 50 Ton/h) and it should not be a substantial increment in the costs. On the other hand, it has to be balanced with the advantage of the PEF technology compared to freeze-thaw since the quality of PEF-treated fish meat is better than the one after the freeze-thaw process.

  1. I agree with the suggestion written by the author for further research on tests that could be carried out on naturally parasitized fish to confirm the potential of PEF as a new strategy for the inactivation of Anisakis in fishery products.

Thank you very much for the comment. This is something that we are now working on and the results are aligned with the ones shown in this manuscript.

Reviewer 2 Report

The article entitled "Evaluation of Pulsed Electric Fields (PEF) parameters in the in-activation of Anisakis larvae in saline solution and hake meat" by Abad et al., is an interesting original article which includes valuable information in the field of the destruction of Anisakis larvae in fresh fish without the need for a freezing process. The introduction provides an adequate theoretical framework, with sufficient bibliography and clearly indicates the objective of this work. Likewise, the material and methods section describes the procedures carried out in a correct way. Moreover, the results and their discussion are also well exposed. 

Abstract

Indicate (PEF) after Pulsed Electric Fields, since later in the summary you use the acronym.

Use the same font throughout the entire abstract.

Introduction

Replace "hours" with "h".

Material and Methods

Replace "hours" with "h".

Replace "Equation 2" with " Equation 1".

Results and Discussion

For a clearer visualization of the moisture, WHC, CL and color I think that these should be expressed in a table like the one in the example. It would also be interesting to include the p-value or, failing that, the significance "n.s.", "*","**", "***" in the last column (headed as Sig.). Also remember to add the letters that indicate the groups and a table footer.

Fresh

PEF

F/ T

SEM

Sig.

Moisture (%)

Water holding capacity (WHC) (%)

Cooking loss (CL) (%)

L* (lightness)

a* (redness)

B* (yellowness)

PEF: PEF-treated (3 kV/ cm, 50 kJ/ kg and 30 µs); F/ T:  frozen/ thawed. SEM: Standar error of the mean. Sig: Significance; *(P< 0.05); **(P< 0.01); ***(P< 0.001); ns: no significant/ or directly the p-value. a-cMeans within the same row not followed by the same letter differ significantly (P< 0.05).

Author Response

The article entitled "Evaluation of Pulsed Electric Fields (PEF) parameters in the in-activation of Anisakis larvae in saline solution and hake meat" by Abad et al., is an interesting original article which includes valuable information in the field of the destruction of Anisakis larvae in fresh fish without the need for a freezing process. The introduction provides an adequate theoretical framework, with sufficient bibliography and clearly indicates the objective of this work. Likewise, the material and methods section describes the procedures carried out in a correct way. Moreover, the results and their discussion are also well exposed. 

Thank you very much for your comments that have been very useful to improve the quality of the manuscript.

Abstract

Indicate (PEF) after Pulsed Electric Fields, since later in the summary you use the acronym.

Following the indications of the reviewer, PEF has been indicated.

Use the same font throughout the entire abstract.

The same font has been used following the suggestions of the reviewer.

Introduction

Replace "hours" with "h".

“hours” has been replaced by “h”.

Material and Methods

Replace "hours" with "h".

“hours” has been replaced by “h”.

Replace "Equation 2" with " Equation 1".

“Equation 2” has been replaced by “Equation 1”.

Results and Discussion

For a clearer visualization of the moisture, WHC, CL and color I think that these should be expressed in a table like the one in the example. It would also be interesting to include the p-value or, failing that, the significance "n.s.", "*","**", "***" in the last column (headed as Sig.). Also remember to add the letters that indicate the groups and a table footer.

Fresh

PEF

F/ T

SEM

Sig.

Moisture (%)

Water holding capacity (WHC) (%)

Cooking loss (CL) (%)

L* (lightness)

a* (redness)

B* (yellowness)

PEF: PEF-treated (3 kV/ cm, 50 kJ/ kg and 30 µs); F/ T:  frozen/ thawed. SEM: Standar error of the mean. Sig: Significance; *(P< 0.05); **(P< 0.01); ***(P< 0.001); ns: no significant/ or directly the p-value. a-cMeans within the same row not followed by the same letter differ significantly (P< 0.05).

Thank you very much for the suggestion. When preparing the manuscript, initially we made a table similar to the one proposed by the reviewer, but after some discussion among the authors, we realized that the plots included in the manuscript enable more easily to compare the results among treatments and to show the statistical differences. On the other hand, following the indications of the reviewer, we have included the significance of the differences.